# Eating Fast Is Associated with Nonalcoholic Fatty Liver Disease in Men But Not in Women with Type 2 Diabetes: A Cross-Sectional Study

**DOI:** 10.3390/nu12082174

**Published:** 2020-07-22

**Authors:** Fuyuko Takahashi, Yoshitaka Hashimoto, Rena Kawano, Ayumi Kaji, Ryosuke Sakai, Yuka Kawate, Takuro Okamura, Emi Ushigome, Noriyuki Kitagawa, Saori Majima, Takafumi Sennmaru, Hiroshi Okada, Naoko Nakanishi, Masahide Hamaguchi, Mai Asano, Masahiro Yamazaki, Michiaki Fukui

**Affiliations:** 1Department of Endocrinology and Metabolism, Graduate School of Medical Science, Kyoto Prefectural University of Medicine, 465 Kajii-cho, Kawaramachi-Hirokoji, Kamigyo-ku, Kyoto 602-8566, Japan; fuyuko-t@koto.kpu-m.ac.jp (F.T.); rena0421@koto.kpu-m.ac.jp (R.K.); kaji-a@koto.kpu-m.ac.jp (A.K.); sakaryo@koto.kpu-m.ac.jp (R.S.); yukawate@koto.kpu-m.ac.jp (Y.K.); d04sm012@koto.kpu-m.ac.jp (T.O.); emis@koto.kpu-m.ac.jp (E.U.); nori-kgw@koto.kpu-m.ac.jp (N.K.); saori-m@koto.kpu-m.ac.jp (S.M.); semmarut@koto.kpu-m.ac.jp (T.S.); conti@koto.kpu-m.ac.jp (H.O.); naoko-n@koto.kpu-m.ac.jp (N.N.); mhama@koto.kpu-m.ac.jp (M.H.); maias@koto.kpu-m.ac.jp (M.A.); masahiro@koto.kpu-m.ac.jp (M.Y.); michiaki@koto.kpu-m.ac.jp (M.F.); 2Department of Diabetology, Kameoka Municipal Hospital, 1-1 Noda, Shinochoshino, Kameoka-city, Kyoto 621-8585, Japan; 3Department of Diabetes and Endocrinology, Matsushita Memorial Hospital, 5-55 Sotojima-cho, Moriguchi 570-8540, Japan

**Keywords:** eating speed, diet, NAFLD, type 2 diabetes

## Abstract

Non-alcoholic fatty liver disease (NAFLD), often complicated by type 2 diabetes mellitus (T2DM), is reported to be associated with diet habits, including eating speed, in the general population. However, the association between eating speed and NAFLD in patients with T2DM, especially sex difference, has not been reported so far. This cross-sectional study included 149 men and 159 women with T2DM. Eating speed was evaluated by a self-reported questionnaire and divided into three groups: fast, moderate, and slow eating. Nutrition status was evaluated by a brief-type self-administered diet history questionnaire. NAFLD was defined as the hepatic steatosis index ≥36 points. Body mass index and carbohydrate/fiber intake in the fast-eating group were higher than those in the slow-eating group in men, whereas this difference was absent in women. In men, compared with eating slowly, eating fast had an elevated risk of the presence of NAFLD after adjusting for covariates (odds ratio (OR) 4.48, 95% confidence interval (CI) 1.09–18.5, *p* = 0.038). In women, this risk was not found, but fiber intake was found to be negatively associated with the presence of NAFLD (OR 0.85, 95% Cl 0.76–0.96, *p* = 0.010). This study indicates that eating speed is associated with the presence of NAFLD in men but not in women.

## 1. Introduction

Type 2 diabetes mellitus (T2DM) has a close relationship with multiple organ abnormalities and various pathophysiological abnormalities, such as metabolic syndrome, non-alcoholic fatty liver disease (NAFLD) [1], arthritis [2], sleep apnea [3] and cardiovascular disease (CVD) [4]. NAFLD is characterized by intrahepatic ectopic fat accumulation due to insulin resistance and abdominal obesity, and the presence of NAFLD correlates with visceral, intramuscular, epicardial, and perivascular fat accumulation [5,6]. The frequency of NAFLD is particularly high in people with T2DM; it has been reported that about 50 to 60% of patients with T2DM have NAFLD [7,8]. Longitudinal studies suggest a strong bidirectional relationship between NAFLD and type 2 diabetes [1].

Lifestyle, including eating habits, is an important factor for NAFLD. Excessive energy intake has been reported to be an important factor in increasing the fat content of the liver [7]. In addition, eating fast is known to be associated with overweight and obesity in adults [9,10,11]. According to previous studies on eating behavior, eating slowly is associated with lower energy intake in comparison to eating fast [12,13]. Eating speed is related to the number of chews; an increased number of chews reduces body weight [9] by increasing satiety [14] and reduces meal consumption [15]. Besides, eating speed is reported to be associated with insulin resistance [16]. Moreover, it has been shown that eating fast is associated with metabolic syndrome [10,17] and T2DM [16,18]. Furthermore, several studies have also revealed that eating fast is associated with NAFLD in the general population [19,20,21]. However, the association between eating speed and NAFLD in patients with T2DM has not been established. Moreover, the influence of the sex of the patient on this association is another important factor, which has not yet been studied. Therefore, we aimed to determine the correlation between eating speed and NAFLD in men and women with T2DM, separately in this cross-sectional study.

## 2. Methods

### 2.1. Study Participants

We are carrying out an ongoing prospective cohort study, we named this cohort study as the KAMOGAWA-DM cohort study, since 2014 to clarify the natural history of people with diabetes [22]. This study included the outpatients of the Department of Endocrinology and Metabolism at Kyoto Prefectural University of Medicine Hospital (Kyoto, Japan) and the Department of Diabetology at Kameoka Municipal Hospital (Kameoka, Japan). In this study, we included patients who answered questionnaires from January 2016 to December 2018. We excluded patients with the following characteristics: viral hepatitis, liver cancer, missing laboratory data, non-T2DM, alcohol ≥20 g/day [23], and incomplete questionnaires. This study was approved by the local research ethics committee (No. RBMR-E-466-5) and was conducted in accordance with the Declaration of Helsinki. Written informed consent was obtained from all the patients.

### 2.2. Lifestyle Characteristics and Measurement

Physical activity and smoking status were evaluated by using the questionnaire. Habit of smoking was defined as current smoker or not. Habit of exercise was defined as performing any kind of physical activity at least once a week. Furthermore, the duration of diabetes was also asked for all patients.

We collected the venous blood after overnight fasting and checked the following factors: hemoglobin A1c (assayed by high-performance liquid chromatography), plasma glucose, total cholesterol, triglycerides gamma-glutamyl transpeptidase, alanine aminotransferase (ALT), and aspartate aminotransferase (AST). 

Body mass index (BMI) was calculated as the weight in kilograms divided by height in meters squared. Ideal body weight (IBW) was calculated as 22 multiplied by the square of patient’s height in meters squared [24]. When the self-reports of dietary intake were performed, blood pressures were measured using a HEM-906 device (OMRON, Kyoto, Japan) in a quiet space after 5 min of rest. The data regarding antidiabetic medication were also obtained from medical records.

### 2.3. Questionnaire for Dietary Habit

Brief-type self-administered diet history questionnaire (BDHQ) was used for evaluating patients’ habitual food and nutrient intake during the previous month. The details and validity of the BDHQ were reported previously [25]. Patient data on energy intake, carbohydrate intake, protein intake, fat intake, fiber intake, and alcohol consumption were collected using the BDHQ. Energy intake (kcal/kg IBW/day) was calculated as dietary total energy (kcal/day) divided by IBW (kg). Carbohydrate/fiber intake was also calculated as carbohydrate intake divided by fiber intake [26].

Eating speed was evaluated by the following statements: very fast, a little fast, normal, a little slow, or very slow, and then we defined “very fast” or “a little fast” as “fast speed eating,” “normal” as “moderate speed eating,” and “very slow” or “a little slow” as “slow speed eating.”

### 2.4. Definition of NAFLD

In this study, we excluded patients who drunk alcohol ≥20 g/day [23]. Hepatic steatosis index (HSI) was used for the presence of NAFLD. The formula of HSI [27] was follows: HSI = 8 × (ALT/AST) + BMI (+2, if impaired fasting glucose (all patients had diabetes in this study); +2, if female). In this study, NAFLD was defined as HSI ≥ 36 points.

### 2.5. Statistical Analysis

We performed statistical analyses using EZR (version 1.40) (Saitama Medical Center, Jichi Medical University, Saitama, Japan) [28], and *p*-value < 0.05 was considered to be statistically significant. Continuous variables were described as the mean (± standard deviation) and categorical variables were described as the number. Since the characteristics and dietary intake were different for men and women, we separately analyzed the patients by sex. 

The differences in characteristics and dietary intake among the patients with different eating speeds were evaluated by the Kruskal-Wallis test and Steel-Dwass test or chi-squared test. 

We then performed logistic regression analyses to examine the effects of eating speed on the presence of NAFLD. The following factors were considered as independent variables: sex, age, duration of diabetes, hemoglobin A1c, insulin treatment, energy intake, carbohydrate intake, dietary fiber intake, habit of exercise, and habit of smoking. BMI is a risk factor for NAFLD [29]. However, BMI was already included the components of HSI; thus, we did not include BMI in the covariates.

## 3. Results

In the present study, 425 patients (234 men and 191 women) were included. We excluded 117 patients: 10 patients had viral hepatitis, 1 patient had liver cancer, 42 patients did not have T2DM, 42 patients had a habit of alcohol consumption, 21 patients did not complete the questionnaires and one patient did not perform laboratory examination; therefore, the final study population included 308 patients (149 men and 159 women) (Figure 1).

The characteristics and dietary intake of study patients are shown in Table 1. Mean age, BMI and HSI were 66.8 ± 10.6 years, 24.2 ± 4.0 kg/m^2^ and 35.3 ± 5.9 points, respectively, in all subjects. The proportion of NAFLD was 36.7% and the proportions of fast, moderate and slow eating were 42.1%, 36.6% and 21.6%, respectively, in all subjects. Furthermore, mean HSI of women (36.4 ± 6.7 points) was higher than that of men (34.2 ± 4.8 points) (*p* = 0.006), and the percentage of the presence of NAFLD of women (43.4%) was higher than that of men (29.5%) (*p* = 0.016). On the other hand, the percentage of patients eating fast was not differ between men and women. In addition, the carbohydrate/fiber intake of men (20.4 ± 7.6) was higher than that of women (17.7 ± 6.0) (*p* < 0.001).

The characteristics of patients according to their eating speed are shown in Table 2. In the case of men, patients eating fast had a higher BMI (*p* = 0.007) and were younger (*p* = 0.009) than the patients eating slowly. Although energy intake (kcal/kg IBW/day) was not different among the groups, carbohydrate/fiber intake of patients eating fast was higher than that of patients eating slowly. The percentages of the presence of NAFLD in patients with fast, moderate, and slow eating speeds were 37.3%, 27.1%, and 11.5%, respectively. However, in women there was no difference in age, BMI, and nutrient parameters among the groups. The percentages of the presence of NAFLD in patients with fast, moderate, and slow eating speeds were 47.1%, 43.8%, and 32.0%, respectively.

The odds ratios (ORs) for the presence of NAFLD across the three categories of eating speeds (fast, moderate, and slow) are shown in Table 3. Compared with eating slowly, easting fast tended to be associated with the presence of NAFLD in both men and women (OR 2.10, 95% 95% confidence interval (CI) 0.90–4.91, *p* = 0.087), although it did not reach statistically significance. Compared with eating slowly, eating fast had an elevated risk of the presence of NAFLD after adjusting for covariates (OR 4.48, 95% CI1.09–18.5, *p* = 0.038) in men. In contrast, there was no relationship between eating speed and the risk of the presence of NAFLD in women. However, fiber intake (OR 0.85, 95% Cl 0.76–0.96, *p* = 0.010) was negatively associated and energy intake (OR 1.05, 95% Cl 1.00–1.10, *p* = 0.044) was positively associated with the presence of fatty liver in women.

## 4. Discussion

This is the first study that investigates the association between eating speed and the presence of NAFLD in patients with T2DM in men and women separately. The results of this study show that eating speed is associated with the presence of NAFLD in men but not in women.

In this study, 47.1% of the participants were categorized as “eating fast”. There is a possibility that patients withT2DM might eat faster than apparently healthy subjects. In fact, a previous study reported that the prevalence of eating fast in patients with diabetes was 61.5% [30], which was higher than that in healthy subjects [16,18]. In addition, in this study, eating fast was associated with NAFLD in men, which is the same as previous studies [19,20,21]. On the other hand, eating speed was not association with the presence of NAFLD in women. This result is a new finding, because the association between NAFLD and eating speed by sex was not evaluated, previously.

Eating fast is known to be associated with overweight and obesity [9,10,11], insulin resistance [16], metabolic syndrome [10,17] and T2DM [16,18]. Eating fast is also reported to be associated with a risk factor for CVD [31], which is associated with NAFLD [4,32].

The possible explanations for the association between eating speed and NAFLD are as follows. The action of chewing results in biting down of the food particles, not only aiding digestion and absorption in the stomach and intestines but also stimulates the satiety center and prevents appetite [14,33]. Chewing well over time also activates the brain, which releases more neuronal histamine and makes the person feel fuller [34]. Neuronal histamine is thought to stimulate the satiety center and sympathetic nerves. Additionally, ghrelin, GLP-1, and peptide YY (PYY) act on the hypothalamus and play a role in the regulation of hunger, satiety, and energy intake [35]. Recently, a study suggested that eating slowly increases the postprandial response of GLP-1 and PYY [35,36]. These results suggest that patients who eat fast may have a lower secretion of GLP-1 and PYY, which may lead to a higher energy intake because they do not feel satiated. Although the difference is not statistically significant, the energy intake in case of patients who ate fast was higher than that of others in men. Hence, it is quite possible that patients who eat fast under-reported their food intake. Since the BDHQ is based on the self-reported frequency of intake, there is a possibility that the answer might not be the actual intake, especially for obese individuals. It has been previously reported that obese individuals tend to under-report their food intake [37]. Moreover, eating slowly might inhibit the secretion of excess insulin because of slower absorption of nutrients. A previous study revealed that the plasma concentrations of insulin were reduced by α-glucosidase inhibition, which slowed down absorption in the intestines [38,39]. Eating fast causes a rapid rise in blood sugar levels and insulin is excessively secreted. Thus, this effect may have facilitated fat accumulation in the liver.

Another possible mechanism is diet-induced thermogenesis (DIT). DIT is the increase in energy expenditure associated with the digestion, absorption, and storage of food and accounts for approximately 10–15% of total daily energy expenditure [40]. Eating slowly is reported to enhance DIT [40,41]. 

Eating speed was associated with carbohydrate/fiber intake in men. Previous studies revealed that carbohydrate/fiber intake is associated with metabolic syndrome [27] and visceral fat accumulation [42]. Thus, there is a possibility that poor diet quality in men with the habit of fast eating is the cause of the presence of NAFLD. In contrast, we did not find an association between eating speed and the presence of NAFLD in women. One of the possible reasons is that eating speed in this study was self-reported, and the difference in eating speed would be small in women. In fact, an earlier study has reported that eating speed is slower in women than in men [43]. Another point to be considered is the diet quality. In this study, energy intake and dietary fiber intake were found to be associated with the presence of NAFLD in women. Excessive energy intake has been reported to be an important factor in increasing the fat content of the liver [7]. Moreover, dietary fiber intake was found to be associated with the presence of NAFLD. Poor dietary fiber intake promotes the development of NAFLD [44]. There was no difference in carbohydrate/fiber intake or dietary fiber intake among the different groups for eating speed in women. These results suggest that women with T2DM were able to consume a better-balanced diet regardless of the eating speed. Consuming a good-quality diet in any group differing by eating speeds may have influenced the lack of an association between eating speed and the presence of NAFLD.

In this study, the presence of NAFLD was associated with HbA1c in men. A previous study reported that HbA1c levels were significantly correlated with NAFLD [45]. A previous study revealed that aging is a risk factor for NAFLD in premenopausal women [46]. In this study, women with NAFLD were significantly younger. This might be because that all of the participants in the study were type 2 diabetes, and the effect of diabetes on NAFLD is greater than that of age.

However, there were certain limitations to our study. First, we did not check the patients’ eating speed directly and depended on answers provided by them. Additionally, the number of chews was not examined in this study. However, a self-reported eatingspeed has been shown to be well-correlated with that reported by a friend or one that is objectively measured [10,11,47]. Second, although liver biopsy is the gold standard for diagnosing NAFLD, we did not perform it in this study. Furthermore, NAFLD was not diagnosed using fibroscan, MRI or ultrasound, although liver ultrasound is still the most convincing one for the diagnosis of fatty liver among non-invasive techniques [48]. However, HSI is correlated with definition of NAFLD using ultrasound [28]. Third, this is a cross-sectional study; thus, we cannot reveal a causal effect. Finally, this study included only Japanese patients; generalization to other ethnic groups is hence uncertain.

In conclusion, to the best of our knowledge, this study showed that eating speed is associated with the presence of NAFLD in men but not in women. Our study indicated that eating habits such as eating fast should be changed; if not, at least, the quality of diet should be improved, such as eating more fiber, which is desirable.

## Figures and Tables

**Figure 1 nutrients-12-02174-f001:**
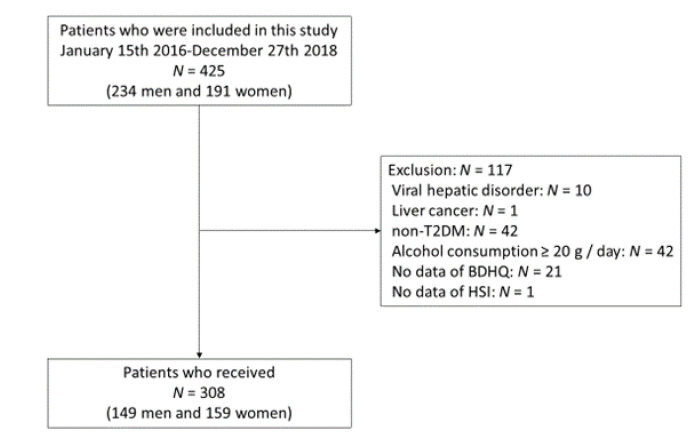
Inclusion and exclusion flow. T2DM, type 2 diabetes mellitus; BDHQ, Brief-type self-administered diet history questionnaire; HSI, hepatic steatosis index.

**Table 1 nutrients-12-02174-t001:** Clinical characteristic of study patients.

	All	Men	Women	*p*
	308	149 (48.4%)	159 (51.6%)	
Age (years)	66.8 (10.6)	67.6 (10.8)	66.1 (10.4)	0.320
BMI (kg/m^2^)	24.2 (4.0)	23.9 (3.0)	24.6 (4.7)	0.475
Duration of diabetes (years)	14.4 (10.2)	15.5 (9.7)	13.3 (10.6)	0.008
SBP (mm Hg)	133.8 (18.8)	132.6 (17.6)	135.0 (19.8)	0.386
DBP (mm Hg)	78.5 (11.3)	78.7 (10.8)	78.3 (11.8)	0.351
Hemoglobin A1c (%)	7.3 (1.2)	7.3 (1.2)	7.3 (1.2)	0.951
Hemoglobin A1c (mmol/mol)	56.5 (13.6)	56.7 (13.6)	56.4 (13.5)	0.951
Glucose (mmol/L)	8.1 (2.6)	8.3 (2.4)	8.0 (2.8)	0.186
Total cholesterol (mmol/L)	5.0 (1.2)	4.8 (0.9)	5.1 (1.3)	<0.001
Triglyceride (mmol/L)	1.5 (0.8)	1.5 (0.9)	1.5 (0.8)	0.701
AST (IU/L)	22.8 (9.6)	24.0 (10.6)	21.7 (8.5)	0.014
ALT (IU/L)	23.3 (14.5)	25.6 (15.3)	21.2 (13.5)	0.001
γ-GTP (IU/L)	32.7 (32.0)	37.0 (40.8)	28.6 (20.1)	0.018
HSI (point)	35.3 (5.9)	34.2 (4.8)	36.4 (6.7)	0.006
Habit of exercise	-	-	-	0.211
No	155 (50.3%)	69 (46.3%)	86 (54.1%)	-
Yes	153 (49.7%)	80 (53.7%)	73 (45.9%)	-
Habit of smoking	-	-	-	0.002
No	270 (87.7%)	121 (81.2%)	149 (93.7%)	-
Yes	38 (12.3%)	28 (18.8%)	10 (6.3%)	-
Insulin treatment	-	-	-	0.827
No	235 (76.3%)	115 (77.2%)	120 (75.5%)	-
Yes	73 (23.7%)	34 (22.8%)	39 (24.5%)	-
Fatty liver				0.016
No	195 (63.3%)	105 (70.5%)	90 (56.6%)	-
Yes	113 (36.7%)	44 (29.5%)	69 (43.4%)	-
Eating speed	-	-	-	0.340
Fast	145 (47.1%)	75 (50.3%)	70 (44.0%)	-
Moderate	112 (36.4%)	48 (32.2%)	64 (40.3%)	-
Slow	51 (16.6%)	26 (17.4%)	25 (15.7%)	-
Total energy intake (kcal/kg IBW/day)	29.8 (10.1)	30.0 (9.0)	29.7 (11.1)	0.426
Protein intake (% Energy)	17.1 (3.4)	16.3 (3.1)	17.9 (3.6)	<0.001
Fat intake (% Energy)	29.2 (6.2)	28.5 (6.3)	29.8 (6.0)	0.022
Carbohydrate intake (% Energy)	52.0 (8.3)	52.9 (8.4)	51.1 (8.2)	0.033
Dietary fiber intake (g/day)	12.3 (4.9)	12.9 (5.1)	11.8 (4.7)	0.047
Carbohydrate/fiber	19.0 (6.9)	20.4 (7.6)	17.7 (6.0)	<0.001

Data are expressed as mean (standard deviation) or number. BMI, body mass index; SBP, systolic blood pressure; DBP, diastolic blood pressure; AST, aspartate aminotransferase; ALT, alanine aminotransferase; γ-GTP, gamma-glutamyl transpeptidase; HSI, hepatic steatosis index. The differences of characteristics between patients by sex were evaluated by Mann-Whitney *u* test or Chi-squared test.

**Table 2 nutrients-12-02174-t002:** The characteristics of the patients according to eating speed.

	Men				Women			
	Fast, *n* = 75 (50.3%)	Moderate, *n* = 48 (32.2%)	Slow, *n* = 26 (17.4%)	*p*	Fast, *n* = 70 (44.0%)	Moderate, *n* = 64 (40.3%)	Slow, *n* = 25 (15.7%)	*p*
Age (years)	66.3 (11.0)	67.1 (10.7)	72.5 (9.4) *	0.012	65.7 (11.1)	65.3 (10.0)	69.2 (9.6)	0.109
BMI (kg/m^2^)	24.3 (2.9)	24.1 (3.0)	22.2 (2.6) *	0.007	25.3 (4.9)	24.2 (4.5)	23.6 (4.7)	0.147
Duration of diabetes (years)	15.6 (10.3)	15.9 (9.2)	14.8 (9.1)	0.891	13.5 (10.2)	12.0 (10.3)	16.0 (12.4)	0.244
SBP (mm Hg)	133.2 (16.3)	134.2 (19.7)	128.2 (17.1)	0.421	137.0 (20.5)	132.5 (19.6)	135.7 (18.1)	0.442
DBP (mm Hg)	78.9 (9.7)	80.5 (12.1)	74.7 (10.6)	0.186	78.9 (12.5)	77.8 (11.4)	77.7 (11.1)	0.886
HbA1c (%)	7.3 (1.2)	7.4 (1.6)	7.3 (0.8)	0.475	7.4 (1.1)	7.3 (1.5)	7.1 (0.7)	0.358
HbA1c (mmol/mol)	56.5 (12.6)	57.1 (17.2)	56.4 (8.6)	0.475	57.4 (11.6)	56.4 (16.9)	54.0 (8.1)	0.358
Glucose (mmol/L)	8.1 (2.4)	8.4 (2.5)	8.4 (2.5)	0.577	8.1 (3.0)	8.0 (2.5)	7.9 (2.7)	0.947
Total cholesterol (mmol/L)	4.8 (0.9)	4.6 (1.0)	4.9 (0.7)	0.135	5.1 (1.4)	5.0 (1.4)	5.4 (0.9)	0.531
Triglyceride (mmol/L)	1.5 (0.8)	1.6 (0.9)	1.3 (0.9)	0.201	1.5 (0.8)	1.5 (1.0)	1.3 (0.6)	0.670
AST (IU/L)	24.5 (12.3)	23.6 (9.6)	23.5 (6.0)	0.716	22.0 (7.1)	22.4 (10.8)	19.3 (3.9)	0.329
ALT (IU/L)	27.2 (17.5)	23.9 (12.6)	24.1 (13.1)	0.762	22.0 (12.3)	22.1 (16.3)	16.4 (5.7)	0.127
γ-GTP (IU/L)	39.5 (50.1)	33.5 (29.1)	36.3 (27.9)	0.722	27.8 (18.7)	31.1 (21.3)	24.2 (20.7)	0.043
HSI (point)	35.0 (5.0)	34.2 (4.3)	32.2 (4.5)	0.047	37.0 (6.3)	36.2 (7.5)	34.5 (5.0)	0.115
Habit of exercise	-	-	-	0.266	-	-	-	0.491
No	34 (45.3%)	26 (54.2%)	9 (34.6%)	-	38 (54.3%)	32 (50.0%)	16 (64.0%)	-
Yes	41 (54.7%)	22 (45.8%)	17 (65.4%)	-	32 (45.7%)	32 (50.0%)	9 (36.0%)	-
Habit of smoking	-	-	-	0.331	-	-	-	0.166
No	52 (82.7%)	36 (75.0%)	23 (88.5%)	-	63 (90.0%)	61 (95.3%)	25 (100.0%)	-
Yes	13 (17.3%)	12 (25.0%)	3 (11.5%)	-	7 (10.0%)	3 (4.7%)	0 (0.0%)	-
Insulin treatment	-	-	-	0.500	-	-	-	0.795
No	58 (77.3%)	39 (81.2%)	18 (69.2%)	-	52 (74.3%)	50 (78.1%)	18 (72.0%)	-
Yes	17 (22.7%)	9 (18.8%)	8 (30.8%)	-	18 (25.7%)	14 (21.9%)	7 (28.0%)	-
Fatty liver	-	-	-	0.041	-	-	-	0.422
No	47 (62.7%)	35 (72.9%)	23 (88.5%)	-	37 (52.9%)	36 (56.2%)	17 (68.0%)	-
Yes	28 (37.3%)	13 (27.1%)	3 (11.5%)	-	33 (47.1%)	28 (43.8%)	8 (32.0%)	-
Total energy intake (kcal/kg IBW/day)	31.2 (9.9)	28.7 (7.5)	28.8 (8.3)	0.402	29.1 (9.3)	29.8 (10.8)	31.2 (16.1)	0.993
Protein intake (% Energy)	15.9 (2.8)	16.6 (3.3)	17.0 (3.2)	0.325	17.6 (3.4)	18.4 (3.9)	17.1 (3.0)	0.312
Fat intake (% Energy)	27.9 (6.4)	29.5 (6.3)	28.7 (6.3)	0.371	30.3 (6.0)	29.6 (5.9)	29.1 (6.4)	0.722
Carbohydrate intake (% Energy)	53.6 (8.4)	52.0 (8.4)	52.1 (8.2)	0.552	51.0 (8.1)	50.6 (8.1)	53.0 (8.6)	0.573
Dietary fiber intake (g/day)	12.6 (5.4)	13.1 (5.4)	13.3 (3.9)	0.453	11.5 (4.7)	11.7 (4.3)	12.9 (5.5)	0.682
Carbohydrate/fiber	22.3 (8.3)	19.0 (6.3)	17.7 (6.4)	0.002	17.8 (6.0)	17.8 (6.0)	17.0 (6.1)	0.924

*, *p* < 0.017 Fast vs. Slow. Data are expressed as mean (standard deviation) or number. BMI, body mass index; SBP, systolic blood pressure; DBP, diastolic blood pressure; HbA1c, hemoglobin A1c; AST, aspartate aminotransferase; ALT, alanine aminotransferase; γ-GTP, gamma-glutamyl transpeptidase; HSI, hepatic steatosis index. The differences of characteristics between patients with different eating speed were evaluated by Kruskal-Wallis test or Chi-squared test.

**Table 3 nutrients-12-02174-t003:** Logistic regression analysis on the presence of fatty liver.

All	Model 1		Model 2		Model 3	
	OR (95% CI)	*p*	OR (95%CI)	*p*	OR (95% CI)	*p*
Eating speed						
Fast	2.64 (1.25–5.56)	0.011	2.13 (0.95–4.81)	0.068	2.10 (0.90–4.91)	0.087
Moderate	2.10 (0.97–4.54)	0.060	1.45 (0.62–3.38)	0.390	1.42 (0.59–3.44)	0.436
Slow	Ref	-	Ref	-	Ref	-
Women	-	-	1.87 (1.08–3.22)	0.025	1.59 (0.90–2.81)	0.108
Age (years)	-	-	0.95 (0.92–0.97)	<0.001	0.95 (0.92–0.98)	<0.001
Duration of diabetes (years)	-	-	0.95 (0.92–0.98)	0.002	0.95 (0.92–0.98)	0.001
Hemoglobin A1c (mmol/mol)	-	-	1.33 (1.04–1.71)	0.022	1.32 (1.03–1.69)	0.027
Energy intake (kcal/kg IBW/day)	-	-	-	-	1.03 (1.00–1.06)	0.079
Carbohydrate intake (% Energy)	-	-	-	-	1.00 (0.97–1.03)	0.987
Dietary fiber intake (g/day)	-	-	-	-	0.91 (0.85–0.98)	0.011
Habit of exercise	-	-	1.10 (0.65–1.87)	0.713	1.13 (0.66–1.94)	0.662
Habit of smoking	-	-	1.47 (0.67–3.26)	0.340	1.37 (0.62–3.05)	0.440
Insulin treatment	-	-	0.71 (0.36–1.38)	0.311	0.77 (0.39–1.53)	0.456
Men	Model 1		Model 2		Model 3	
	OR (95% CI)	*p*	OR (95%CI)	*p*	OR (95% CI)	*p*
Eating speed						
Fast	4.57 (1.26–16.6)	0.021	4.34 (1.08–17.4)	0.038	4.48 (1.09–18.5)	0.038
Moderate	2.85 (0.73–11.1)	0.132	2.73 (0.61–12.1)	0.186	2.97 (0.66–13.4)	0.156
Slow	Ref	-	Ref	-	Ref	-
Age (years)	-	-	0.96 (0.92–1.00)	0.045	0.96 (0.92–1.00)	0.063
Duration of diabetes (years)	-	-	0.94 (0.89–0.99)	0.017	0.93 (0.88–0.99)	0.014
Hemoglobin A1c (mmol/mol)	-	-	1.14 (0.43–3.02)	0.798	1.04 (1.00–1.08)	0.031
Energy intake (kcal/kg IBW/day)	-	-	-	-	1.00 (0.95–1.06)	0.865
Carbohydrate intake (% Energy)	-	-	-	-	0.99 (0.94–1.04)	0.558
Dietary fiber intake (g/day)	-	-	-	-	0.95 (0.86–1.05)	0.351
Habit of exercise	-	-	2.05 (0.89–4.76)	0.093	2.09 (0.88–4.93)	0.094
Habit of smoking	-	-	1.14 (0.43–3.02)	0.798	1.04 (0.39–2.79)	0.940
Insulin treatment	-	-	0.51 (0.18–1.51)	0.225	0.56 (0.18–1.69)	0.300
Women	Model 1		Model 2		Model 3	
	OR (95% CI)	*p*	OR (95%CI)	*p*	OR (95% CI)	*p*
Eating speed						
Fast	1.90 (0.72–4.96)	0.193	1.33 (0.45–3.95)	0.606	1.30 (0.39–4.31)	0.665
Moderate	1.65 (0.62–4.38)	0.312	1.07 (0.35–3.23)	0.906	0.92 (0.28–3.09)	0.896
Slow	Ref	-	Ref	-	Ref	-
Age (years)	-	-	0.93 (0.90–0.97)	0.001	0.94 (0.90–0.98)	0.006
Duration of diabetes (years)	-	-	0.95 (0.91–0.99)	0.025	0.95 (0.90–0.99)	0.018
Hemoglobin A1c (mmol/mol)	-	-	1.01 (0.98–1.04)	0.429	1.01 (0.98–1.04)	0.513
Energy intake (kcal/kg IBW/day)	-	-	-	-	1.05 (1.00–1.09)	0.044
Carbohydrate intake (% Energy)	-	-	-	-	1.01 (0.96–1.06)	0.743
Dietary fiber intake (g/day)	-	-	-	-	0.85 (0.76–0.96)	0.010
Habit of exercise	-	-	0.63 (0.31–1.31)	0.216	0.65 (0.30–1.40)	0.272
Habit of smoking	-	-	3.08 (0.64–14.8)	0.160	3.30 (0.67–16.2)	0.142
Insulin treatment	-	-	0.91 (0.36–2.27)	0.836	0.92 (0.35–2.40)	0.868

All, Model 1 is unadjusted; Model 2 is adjusted for sex, age, duration of diabetes, hemoglobin A1c, habit of exercise, habit of smoking, insulin treatment; and Model 3 is adjusted for sex, age, duration of diabetes, hemoglobin A1c, energy intake, carbohydrate intake, dietary fiber intake, habit of exercise, habit of smoking, insulin treatment. Men and women, Model 1 is unadjusted; Model 2 is adjusted for age, duration of diabetes, hemoglobin A1c, habit of exercise, habit of smoking, insulin treatment; and Model 3 is adjusted for age, duration of diabetes, hemoglobin A1c, energy intake, carbohydrate intake, dietary fiber intake, habit of exercise, habit of smoking, insulin treatment.

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
