# Peer review of "Eating Fast Is Associated with Nonalcoholic Fatty Liver Disease in Men But Not in Women with Type 2 Diabetes: A Cross-Sectional Study"

_nutrients, 2020, doi:10.3390/nu12082174_

Round 1
Reviewer 1 Report
This study aimed to determine the relationship between eating speed and risk of developing NAFLD and a possible sex difference in this relationship in type 2 diabetic patients. Major findings are that BMI and carbohydrate/fiber intake in the fast-eating group were higher than those in the other groups in men, whereas this difference was absent in women. In men, compared with eating slowly, eating fast had an elevated risk of the presence of NAFLD after adjusting for covariates. In women, this risk was not found, but fiber intake was found to be negatively associated with the presence of NAFLD. Authors concluded that eating speed is associated with the presence of NAFLD in men but not in women. The topic of the present manuscript is interesting. My major concern is that “eating speed”, the major variable of the present study, is based on BDHQ. This is a significant limitation of the present analysis.

Major points
- Definition of eating speed is unclear. What are the specific criteria for each eating speed? Although BDHQ is widely used, since “eating speed” is the major variable of the present study, clear and scientifically-sound definition of eating speed is necessary. Why more patients are categorized as “eating fast” (46.4%) than “normal” (37.4%)? It is difficult to draw the conclusion without a clear definition of “normal” eating speed.
- L140-144: This part of “Results” is not effective and is not clear exactly what point authors want to make here. Only part of Table 1 is described.
- Comparison with previous findings in general population (refs 15 and 16) needs to be discussed (similarity and/or any differences).
Minor point
- L47: …cardiovascular disease, and “what”?
2. Table 1: Sum of fatty liver No (n=189) and Yes (n=112) does not match the total number of subjects (n=302). Why?
Author Response
At first, we would like to thank reviewers for constructive comments on our manuscript. According to your comments, we have carefully revised our manuscript. Responses to your comments are described below.
Response to Reviewer 1
Major
- Definition of eating speed is unclear. What are the specific criteria for each eating speed? Although BDHQ is widely used, since “eating speed” is the major variable of the present study, clear and scientifically-sound definition of eating speed is necessary. Why more patients are categorized as “eating fast” (46.4%) than “normal” (37.4%)? It is difficult to draw the conclusion without a clear definition of “normal” eating speed.
Response
Thank you for your comment. We asked participants to report their eating speed in one of five qualitative categories - very slow, a little slow, moderate, a little fast, or very fast - using a self-reported BDHQ. This method is dependent on the participants’ reports. However, a self-reported eating rate has been shown to be well correlated with that reported by a friend or one that is objectively measured. Thus, we have mentioned this point as one of the limitations of this study in the Discussion section described as below.
Discussion
“First, we did not check the patients’ eating speed directly and depended on answers provided by them.”
“However, a self-reported eating rate has been shown to be well correlated with that reported by a friend or one that is objectively measured [11,12,52].”
Reference
- Ekuni D, Furuta M, Takeuchi N, Tomofuji T, Morita M. Self-reports of eating quickly are related to a decreased number of chews until first swallow, total number of chews, and total duration of chewing in young people. Arch Oral Biol. 2012;57(7):981-986.
Moreover, there was an error in the way of excluding participants. Therefore, we have reanalyzed, and 47.1% of the participants were categorized as "eating fast". There is a possibility that patients with type 2 diabetes might eat faster than apparently healthy subjects. In fact, a previous study reported that the prevalence of eating fast in patients with diabetes was 61.5%, which was higher than that in healthy subjects (50.8% among middle-aged Japanese healthy subjects and 36.5% among 18-year-old healthy Japanese women). According to your comment, we have mentioned this point in the Discussion section described as below.
Discussion
“In this study, 47.1% of the participants were categorized as “eating fast”. There is a possibility that patients with type 2 diabetes might eat faster than apparently healthy subjects. In fact, a previous study reported that the prevalence of eating fast in patients with diabetes was 61.5% [33], which was higher than that in healthy subjects [20,21]. In addition, in this study, eating fast was associated with NAFLD in men, which is the same as previous studies [22-24].”
Reference
- Saito A, Kawai K, Yanagisawa M, Yokoyama H, Kuribayashi N, Sugimoto H, Oishi M, Wada T, Iwasaki K, Kanatsuka A, Yagi N, Okuguchi F, Miyazawa K, Arai K, Saito K, Sone H. Self-reported rate of eating is significantly associated with body mass index in Japanese patients with type 2 diabetes. Japan Diabetes Clinical Data Management Study Group (JDDM26). Appetite. 2012;59(2):252-255.
- L140-144: This part of “Results” is not effective and is not clear exactly what point authors want to make here. Only part of Table 1 is described.
Response
Thank you for your comment. According to your comment, we have revised the Results section described as below.
“The characteristics and dietary intake of study patients are shown in Table 1. Mean age, BMI and HSI were 66.8 ± 10.6 years, 24.2 ± 4.0 kg/m2 and 35.3 ± 5.9 points, respectively, in all subjects. The proportion of NAFLD were 36.7 % and the proportion of fast, moderate and slow eating were 42.1 %, 36.6 % and 21.6 %, respectively, in all subjects. Furthermore, mean HSI of women (36.4 ± 6.7 points) were higher than that of men (34.2 ± 4.8 points) (p = 0.006), and the percentage of the presence of NAFLD of women (43.4 %) was higher than that of men (29.5 %) (p = 0.016). On the other hand, the percentage of patients eating fast was not differ between men and women. In addition, the carbohydrate/ fiber intake of men (20.4 ± 7.6) was higher than that of women (17.7 ± 6.0) (p < 0.001).”
- Comparison with previous findings in general population (refs 15 and 16) needs to be discussed (similarity and/or any differences).
Response
Thank you for your valuable suggestion. The influence of the sex of the patient on this association is an important factor, which has not yet been studied. In this study, eating fast was associated with NAFLD in men, which is the same as previous studies. On the other hand, eating speed was not association with the presence of NAFLD in women. This is a new finding, because the association between NAFLD and eating speed by sex was not evaluated, previously. According to your suggestion, we have added this point in the Discussion section described as below.
“In addition, in this study, eating fast was associated with NAFLD in men, which is the same as previous studies [22-24]. On the other hand, eating speed was not association with the presence of NAFLD in women. This result is a new finding, because the association between NAFLD and eating speed by sex was not evaluated, previously.”
Minor
- L47: …cardiovascular disease, and “what”?
Response
Thank you for your comment. We have revised this sentence as below.
“Type 2 diabetes mellitus (T2DM) has a close relationship with multiple organ abnormalities and various pathophysiological abnormalities, such as metabolic syndrome, non-alcoholic fatty liver disease (NAFLD) [1], arthritis [2], sleep apnea [3] and cardiovascular disease (CVD) [4].”
- Table 1: Sum of fatty liver No (n=189) and Yes (n=112) does not match the total number of subjects (n=302). Why?
Response
Thank you for your comment. We forgot to exclude the patients with missing data of covariates. According to your comment, Figure 1 and all tables have been updated
Reviewer 2 Report
Introduction
There are some typo and grammatical errors that can be corrected
Line 47
Line 59 needs a reference
Method
Line 90 needs an additional coma
Line 93-94 grammar
Definition of NAFLD- using HSI. Were any of the patients diagnosed of having NAFLD by hematologist using fibroscan or MRI or ultrasound?
Statistical analysis: why was a p-value<0.05 used? There were multiple variables tested, why wasn't a multiple testing correction applied?
Results
Model 2 was not significant but model 1 was significant - for associating eating fast with NAFLD in men?
Why log of duration of diabetes used for the logistic regression. Duration of diabetes was associated with NAFLD as well in both men and women, maybe a stronger predictor than eating speed
Even, HbA1c was associated with NAFLD in men
Age was associated with NAFLD in women
Why was these associations not discussed?
I think it will be crucial to include both male and female and perform another set of logistic regressions.
Discussion
Some minor grammatical corrections.
They included good literature discussing about the mechanism how eating fast and/or slow may indirectly influence your diet, satiety and hormones.
Author Response
At first, we would like to thank reviewers for constructive comments on our manuscript. According to your comments, we have carefully revised our manuscript. Responses to your comments are described below.
Response to Reviewer 2
Introduction
There are some typo and grammatical errors that can be corrected
Line 47 
Line 59 needs a reference
Response
Thank you for your comment. Typos and grammatical errors have been corrected and we have added the following references.
Introduction
“Type 2 diabetes mellitus (T2DM) has a close relationship with multiple organ abnormalities and various pathophysiological abnormalities, such as metabolic syndrome, non-alcoholic fatty liver disease (NAFLD) [1], arthritis [2], sleep apnea [3] and cardiovascular disease (CVD) [4].”
References
- Ballestri S, Zona S, Targher G, Romagnoli D, Baldelli E, Nascimbeni F, Roverato A, Guaraldi G, Lonardo A. Nonalcoholic fatty liver disease is associated with an almost twofold increased risk of incident type 2 diabetes and metabolic syndrome. Evidence from a systematic review and meta-analysis. J Gastroenterol Hepatol. 2016;31(5):936-944.
- Dong Q, Liu H, Yang D, Zhang Y. Diabetes mellitus and arthritis: is it a risk factor or comorbidity?: A systematic review and meta-analysis. Medicine (Baltimore). 2017;96(18):e6627.
- Tasali E, Mokhlesi B, Van Cauter E. Obstructive sleep apnea and type 2 diabetes: interacting epidemics. Chest. 2008;133(2):496-506.
- Cai J, Zhang XJ, Ji YX, Zhang P, She ZG, Li H. Nonalcoholic Fatty Liver Disease Pandemic Fuels the Upsurge in Cardiovascular Diseases. Circ Res. 2020;126(5):679-704.
Definition of NAFLD- using HSI. Were any of the patients diagnosed of having NAFLD by hematologist using fibroscan or MRI or ultrasound?
Response
Thank you for your comment. As you say, definition of NAFLD using fibroscan or MRI or ultrasound is important. Unfortunately, however, we did not have those data. Thus, we have mentioned this point as one of the limitations of this study in the Discussion section described as below.
Discussion
“Second, although liver biopsy is the gold standard for diagnosing NAFLD, we did not perform it in this study. Furthermore, NAFLD was not diagnosed using fibroscan, MRI or ultrasound, although liver ultrasound is still the most convincing one for the diagnosis of fatty liver among non-invasive technique [52]. However, HSI is correlated with definition of NAFLD using ultrasound [30].”
Reference
- Ballestri S, Nascimbeni F, Baldelli E, Marrazzo A, Romagnoli D, Targher G, Lonardo A. Ultrasonographic fatty liver indicator detects mild steatosis and correlates with metabolic/histological parameters in various liver diseases. Metabolism. 2017;72:57-65.
Statistical analysis: why was a p-value<0.05 used? There were multiple variables tested, why wasn't a multiple testing correction applied?
Response
Thank you for your valuable suggestion. According to your suggestion, we have reanalyzed and applied a p-value <0.017 for a multiple testing correction.
Results
Model 2 was not significant but model 1 was significant - for associating eating fast with NAFLD in men?
Response
Thank you for your comment. There was an error in the way of excluding participants. We have reanalyzed and found that in each model, compared with eating slowly, eating fast had an elevated risk of the presence of NAFLD after adjusting for covariates in men, in each model. According to your comment, we have revised Table 3.
Why log of duration of diabetes used for the logistic regression. Duration of diabetes was associated with NAFLD as well in both men and women, maybe a stronger predictor than eating speed
Response
Thank you for your comment. According to your comment, we have reanalyzed using not log duration of diabetes but duration of diabetes. Even after adjusting for duration of diabetes, the results did not change.
Even, HbA1c was associated with NAFLD in men
Age was associated with NAFLD in women
Why was these associations not discussed?
Response
Thank you for your comment. As you say, the presence of fatty liver was also affected by HbA1c in this study. A previous study revealed that HbA1c levels have been shown to be significantly correlated with NAFLD. Furthermore, as you say, it is well known that age was associated with NAFLD in premenopausal women. In this study, women with fatty liver were significantly younger. This might be because that all of the participants in the study were type 2 diabetes, and the effect of diabetes on NAFLD is greater than that of age. According to your comment, we have added the following to our discussion.
Discussion
“In this study, the presence of NAFLD was associated with HbA1c in men. A previous study revealed that HbA1c levels have been shown to be significantly correlated with NAFLD [49]. A previous study revealed that aging is a risk factor for NAFLD in premenopausal women [50]. In this study, women with NAFLD were significantly younger. This might be because that all of the participants in the study were type 2 diabetes, and the effect of diabetes on NAFLD is greater than that of age.”
References
- Ma H, Xu C, Xu L, Yu C, Miao M, Li Y. Independent association of HbA1c and nonalcoholic fatty liver disease in an elderly Chinese population. BMC Gastroenterol. 2013;13:3.
- Hamaguchi M, Kojima T, Ohbora A, Takeda N, Fukui M, Kato T. Aging is a risk factor of nonalcoholic fatty liver disease in premenopausal women. World J Gastroenterol. 2012;18(3):237-243.
I think it will be crucial to include both male and female and perform another set of logistic regressions.
Response
Thank you for your valuable comment. According to your comment, we have conducted an analysis, which includes both men and women. Compared with eating slow, easting fast tended to be associated with the prevalence NAFLD (OR 2.10, 95% CI 0.90-4.91, p = 0.087), although it did not reach statistically significance (Table 3). According to your comment, we have added this point in the Results section described as below.
“Compared with eating slowly, easting fast tended to be associated with the prevalence NAFLD in both men and women (OR 2.10, 95% CI 0.90-4.91, p = 0.087), although it did not reach statistically significance.”
Discussion
Some minor grammatical corrections.
They included good literature discussing about the mechanism how eating fast and/or slow may indirectly influence your diet, satiety and hormones.
Response
Thank you for your comment. We have done some grammatical connections.
Reviewer 3 Report
GENERAL COMMENT
The Authors performed an interesting study on the association between eating speed and NAFLD in patients with type 2 diabetes. However, some comments may be raised.
SPECIFIC COMMENTS
- Methods. It is not clear if the how the eating speed was evaluated. Was it assessed by a validated questionnaire or not?
- Methods/Results. Eating fast was not associated with NAFLD in men with T2D at univariate analys but it was at multivariate analysis. Why BMI was not included as covariate in the mutivariate analysis? Please clarify and discuss.
- The Authors state among study limitations that they did not perform liver biopsy. Among non-invasive technique liver ultrasound is still the most convincing one for the diagnosis of fatty liver. All hepatological guidelines consider liver ultrasound as the first-line imaging technique for the diagnosis of fatty liver in both clinical and epidemiological settings due to its safety, cost-effective profile and availability. Recent data suggest that ultrasound can detect steatosis as low as 10-20% and performed better than surrogate scores such as fatty liver index (Metabolism. 2017 Jul;72:57-65. doi: 10.1016/j.metabol.2017.04.003). Therefore, the fact they did not perform ultrasound rather than liver biopsy should be stated as a study limitation in an epidemiological study.
- Introduction. Longitudinal studies suggest a strong bidirectional relationship between NAFLD and type 2 diabetes (e.g. J Gastroenterol Hepatol. 2016 May;31(5):936-44.). Please specify that type 2 diabetes can be both cause and consequence of NAFLD.
- Literature on the relationship between eating speed and NAFLD should be updated:
e.g. Cao X et al. Association between eating speed and newly diagnosed non-alcoholic fatty liver disease among the general population. Nutrition Research, 2020. https://www.sciencedirect.com/science/article/pii/S0271531720304747
- NAFLD is a global epidemic condition independently associated with relevant CVD complications. Literature about the relationship between NAFLD and cardiovascular disease should be updated (Circ Res. 2020;126:679–704; Adv Ther. 2020 May;37(5):1910-1932. doi: 10.1007/s12325-020-01307-z).
- The relationship between eating speed and CVD should be discussed.
E.g. Paz-Graniel I, Babio N, Mendez I, Salas-Salvadó J. Association between Eating Speed and Classical Cardiovascular Risk Factors: A Cross-Sectional Study. Nutrients. 2019;11(1):83. Published 2019 Jan 4. doi:10.3390/nu11010083
Author Response
At first, we would like to thank reviewers for constructive comments on our manuscript. According to your comments, we have carefully revised our manuscript. Responses to your comments are described below.
Response to Reviewer 3
- Methods. It is not clear if the how the eating speed was evaluated. Was it assessed by a validated questionnaire or not?
Response
Thank you for your comment. We asked participants to report their eating speed in one of five qualitative categories - very slow, a little slow, moderate, a little fast, or very fast - using a self-reported BDHQ. This method is dependent on the participants’ reports. However, a self-reported eating rate has been shown to be well correlated with that reported by a friend or one that is objectively measured. Thus, we have mentioned this point as one of the limitations of this study in the Discussion section described as below.
Discussion
“First, we did not check the patients’ eating speed directly and depended on answers provided by them.”
“However, a self-reported eating rate has been shown to be well correlated with that reported by a friend or one that is objectively measured [11,12,51].”
Reference
- Ekuni D, Furuta M, Takeuchi N, Tomofuji T, Morita M. Self-reports of eating quickly are related to a decreased number of chews until first swallow, total number of chews, and total duration of chewing in young people. Arch Oral Biol. 2012;57(7):981-986.
- Methods/Results. Eating fast was not associated with NAFLD in men with T2D at univariate analysis but it was at multivariate analysis. Why BMI was not included as covariate in the multivariate analysis? Please clarify and discuss.
Response
Thank you for your comment. There was an error in the way of excluding participants. Therefore, we have reanalyzed data and have found an association between eating speed and the presence of fatty liver in men at both univariate analysis and multivariate analysis.
BMI is one of the important risk markers of NAFLD. In this study, the presence or absence of fatty liver was determined using HSI. Since BMI was already included as a component of HSI, we did not include BMI in the covariates. According to your comment, we have added this point in the Methods section described as below.
Methods
“BMI is a risk factor for NAFLD [32]. However, BMI was already included to a component of HSI; thus, we did not include BMI in the covariates.”
Reference
- Hashimoto Y, Hamaguchi M, Fukuda T, Nakamura N, Ohbora A, Kojima T, Fukui M. BMI history and risk of incident fatty liver: a population-based large-scale cohort study. Eur J Gastroenterol Hepatol. 2016;28(10):1188-1193.
- The Authors state among study limitations that they did not perform liver biopsy. Among non-invasive technique liver ultrasound is still the most convincing one for the diagnosis of fatty liver. All hepatological guidelines consider liver ultrasound as the first-line imaging technique for the diagnosis of fatty liver in both clinical and epidemiological settings due to its safety, cost-effective profile and availability. Recent data suggest that ultrasound can detect steatosis as low as 10-20% and performed better than surrogate scores such as fatty liver index (Metabolism. 2017 Jul;72:57-65. doi: 10.1016/j.metabol.2017.04.003). Therefore, the fact they did not perform ultrasound rather than liver biopsy should be stated as a study limitation in an epidemiological study.
Response
Thank you for your suggestion. As you say, all hepatological guidelines consider liver ultrasound as the first-line imaging technique and ultrasound can detect steatosis as low as 10-20% and performed better than surrogate scores such as fatty liver index. Unfortunately, however, we did not have data of liver ultrasound. Thus, we have mentioned this point as a one of the limitations of this study in the Discussion section described as below.
Discussion
“Second, although liver biopsy is the gold standard for diagnosing NAFLD, we did not perform it in this study. Furthermore, NAFLD was not diagnosed using fibroscan, MRI or ultrasound, although liver ultrasound is still the most convincing one for the diagnosis of fatty liver among non-invasive technique [52]. However, HSI is correlated with definition of NAFLD using ultrasound [30].”
Reference
- Ballestri S, Nascimbeni F, Baldelli E, Marrazzo A, Romagnoli D, Targher G, Lonardo A. Ultrasonographic fatty liver indicator detects mild steatosis and correlates with metabolic/histological parameters in various liver diseases. Metabolism. 2017;72:57-65.
- Introduction. Longitudinal studies suggest a strong bidirectional relationship between NAFLD and type 2 diabetes (e.g. J Gastroenterol Hepatol. 2016 May;31(5):936-44.). Please specify that type 2 diabetes can be both cause and consequence of NAFLD.
Response
Thank you for your valuable comment. As you say, there was a strong bidirectional relationship between NAFLD and type 2 diabetes. According to your comment, we have added this point in the Introduction section described as below.
Introduction
“NAFLD is characterized by intrahepatic ectopic fat accumulation due to insulin resistance and abdominal obesity, and the presence of NAFLD correlates with visceral, intramuscular, epicardial, and perivascular fat accumulation [5,6]. The frequency of NAFLD is particularly high in people with T2DM; it has been reported that about 50 to 60% of patients with T2DM have NAFLD [7,8]. A longitudinal study suggests a strong bidirectional relationship between NAFLD and type 2 diabetes [9].”
Reference
- Ballestri S, Zona S, Targher G, Romagnoli D, Baldelli E, Nascimbeni F, Roverato A, Guaraldi G, Lonardo A. Nonalcoholic fatty liver disease is associated with an almost twofold increased risk of incident type 2 diabetes and metabolic syndrome. Evidence from a systematic review and meta-analysis. J Gastroenterol Hepatol. 2016;31(5):936-944.
- Literature on the relationship between eating speed and NAFLD should be updated:
e.g. Cao X et al. Association between eating speed and newly diagnosed non-alcoholic fatty liver disease among the general population. Nutrition Research, 2020.
- NAFLD is a global epidemic condition independently associated with relevant CVD complications. Literature about the relationship between NAFLD and cardiovascular disease should be updated (Circ Res. 2020;126:679–704; Adv Ther. 2020 May;37(5):1910-1932. doi: 10.1007/s12325-020-01307-z).
- The relationship between eating speed and CVD should be discussed.
E.g. Paz-Graniel I, Babio N, Mendez I, Salas-Salvadó J. Association between Eating Speed and Classical Cardiovascular Risk Factors: A Cross-Sectional Study. Nutrients. 2019;11(1):83. Published 2019 Jan 4. doi:10.3390/nu11010083
Response
Thank you for your valuable suggestion. According to your suggestion, we have updated literature on the relationship between NAFLD and eating speed and the relationship between NAFLD and CVD. In addition, we have also discussed the relationship between eating speed and CVD in the Introduction and Discussion sections described as below.
Introduction
“Furthermore, several studies have revealed that eating fast is associated with NAFLD in the general population [22-24]”
Discussion
“Eating fast is also reported to be associated with a risk factor for CVD [34], which are associated with NAFLD [35,36].”
References
- Cao X, Gu Y, Bian S, et al. Association between eating speed and newly diagnosed non-alcoholic fatty liver disease among the general population. Nutrition Research. 2020 in press. DOI: 10.1016/j.nutres.2020.06.012
- Paz-Graniel I, Babio N, Mendez I, Salas-Salvadó J. Association between Eating Speed and Classical Cardiovascular Risk Factors: A Cross-Sectional Study. Nutrients. 2019;11(1):83.
- Cai J, Zhang XJ, Ji YX, Zhang P, She ZG, Li H. Nonalcoholic Fatty Liver Disease Pandemic Fuels the Upsurge in Cardiovascular Diseases. Circ Res. 2020;126(5):679-704.
- Ballestri S, Capitelli M, Fontana MC, Arioli D, Romagnoli E, Graziosi C, Lonardo A, Marietta M, Dentali F, Cioni G. Direct Oral Anticoagulants in Patients with Liver Disease in the Era of Non-Alcoholic Fatty Liver Disease Global Epidemic: A Narrative Review. Adv Ther. 2020;37(5):1910-1932.
Round 2
Reviewer 1 Report
The authors have been responsive to my comments and the manuscript has been improved. I have questions for experimental design that need to be clarified.
1) Why exclusion criteria have been changed from previous version? (L78-80)
2) Inconsistent sample sizes are reported.
In the abstract, sample size is 146 men and 156 women. But, these numbers are not inconsistent with those found in tables (149 men and 159 women). Additionally, according to the text (L135-139) and figure 1, exclusion should be 92, not 117. If so, the final sample size should be 334, not 308.
Author Response
At first, we would like to thank reviewers for constructive comments on our manuscript. According to your comments, we have carefully revised our manuscript. Responses to your comments are described below.
- Why exclusion criteria have been changed from previous version? (L78-80)
Response
Thank you for your comment. During the revision process, we have found the errors in handling of the data and exclusion criteria. For details, the participants who took corticosteroids were already included in non-T2DM, and the definition of alcohol used was incorrect. In addition, we have found a patient with active liver cancer. Thus, we have changed the exclusion criteria described as below.
“We excluded patients with the following characteristics: viral hepatitis, liver cancer, missing data of laboratory data, non-T2DM, alcohol ≥20 g/day [23], and incomplete questionnaires.”
- Inconsistent sample sizes are reported.
In the abstract, sample size is 146 men and 156 women. But, these numbers are not inconsistent with those found in tables (149 men and 159 women). Additionally, according to the text (L135-139) and figure 1, exclusion should be 92, not 117. If so, the final sample size should be 334, not 308.
Response
We’re very sorry to confuse you. We have rechecked and revised it.
Abstract
“This cross-sectional study included 149 men and 159 women with T2DM.”
Results
“In the present study, 425 patients (234 men and 191 women) were included. We excluded 117 patients: 10 patients had viral hepatitis, one patient had liver cancer, 42 patients did not have T2DM, 42 patients had a habit of alcohol consumption, 21 patients did not complete the questionnaires and one patient did not perform laboratory examination; therefore, the final study population included 308 patients (149 men and 159 women).”
Reviewer 3 Report
The study has improved. Please find some other comments.
Reference 1 and 9 are duplicated.
Line 52. "A longitudinal study suggests... [9]." Please change to "Longitudinal studies suggest ...".
English should be revised: e.g.
- BMI was already included to a component.
- The proportion of NAFLD were 36.7 %... Furthermore, mean HSI of women (36.4 ± 6.7 points) were higher than that of men (34.2 ± 4.8 points) (p = 0.006)...
- Eating fast is also reported to be associated with a risk factor for CVD [34], which are associated with NAFLD [35,36].
- A previous study revealed that HbA1c levels have been shown to be significantly correlated with NAFLD
Author Response
At first, we would like to thank reviewers for constructive comments on our manuscript. According to your comments, we have carefully revised our manuscript. Responses to your comments are described below.
- Reference 1 and 9 are duplicated.
Response
Thank you for your comment. We have corrected the duplicate references.
- Line 52. "A longitudinal study suggests... [9]." Please change to "Longitudinal studies suggest ...".
Response
Thank you for your comment. According to your comment, we have revised the Introduction section described as below.
“Longitudinal studies suggest a strong bidirectional relationship between NAFLD and type 2 diabetes [1].”
- English should be revised: e.g.
- BMI was already included toa component.
- The proportion of NAFLD were7 %... Furthermore, mean HSI of women (36.4 ± 6.7 points) werehigher than that of men (34.2 ± 4.8 points) (p = 0.006)...
- Eating fast is also reported to be associated with a risk factor for CVD [34], which areassociated with NAFLD [35,36].
- A previous study revealed thatHbA1c levels have been shown to be significantly correlated with NAFLD
Response
Thank you for your comment. We have revised these sentences as below.
“However, BMI was already included the components of HSI; thus, we did not include BMI in the covariates.”
“The proportion of NAFLD was 36.7 % and the proportions of fast, moderate and slow eating were 42.1 %, 36.6 % and 21.6 %, respectively, in all subjects. Furthermore, mean HSI of women (36.4 ± 6.7 points) was higher than that of men (34.2 ± 4.8 points) (p = 0.006), and the percentage of the presence of NAFLD of women (43.4 %) was higher than that of men (29.5 %) (p = 0.016).”
“Eating fast is also reported to be associated with a risk factor for CVD [31], which is associated with NAFLD [32,33].”
“A previous study reported that HbA1c levels were significantly correlated with NAFLD [46].”